Association detection between multiple traits and rare variants based on family data via a nonparametric method

Chi Jinling 1 2
Xu Meijuan 1
Sheng Xiaona 3
Zhou Ying zhouying@hlju.edu.cn 1
1 Department of Statistics, Heilongjiang University , Harbin , China
2 School of Mathematics and Statistics, Xidian University , Xi’an , China
3 School of Information Engineering, Harbin University , Harbin , China
Pfeffer Ulrich
Electronic publication date: 2023 Sep 26
Publication date: 2023
Volume: 11
Electronic Location ID: e16040
Received 2023 Mar 20; Accepted 2023 Aug 15
Copyright: ©2023 Chi et al.
Copyright year: 2023
Copyright holder: Chi et al.
License: This is an open access article distributed under the terms of the Creative Commons Attribution License, which permits unrestricted use, distribution, reproduction and adaptation in any medium and for any purpose provided that it is properly attributed. For attribution, the original author(s), title, publication source (PeerJ) and either DOI or URL of the article must be cited.
License URL: https://creativecommons.org/licenses/by/4.0/

Keywords: Family-based design, Multiple traits, Rare variants, The generalized Kendall’s τ

Funding: The National Natural Science Foundation of China 12071114 This research was supported by the National Natural Science Foundation of China (Grant No. 12071114). The funders had no role in study design, data collection and analysis, decision to publish, or preparation of the manuscript.

==============================
Background

The rapid development of next-generation sequencing technologies allow people to analyze human complex diseases at the molecular level. It has been shown that rare variants play important roles for human diseases besides common variants. Thus, effective statistical methods need to be proposed to test for the associations between traits (e.g., diseases) and rare variants. Currently, more and more rare genetic variants are being detected throughout the human genome, which demonstrates the possibility to study rare variants. Yet complex diseases are usually measured as a variety of forms, such as binary, ordinal, quantitative, or some mixture of them. Therefore, the genetic mapping problem can be attributable to the association detection between multiple traits and multiple loci, with sufficiently considering the correlated structure among multiple traits.

Methods

In this article, we construct a new non-parametric statistic by the generalized Kendall’s τ theory based on family data. The new test statistic has an asymptotic distribution, it can be used to study the associations between multiple traits and rare variants, which broadens the way to identify genetic factors of human complex diseases.

Results

We apply our method (called Nonp-FAM) to analyze simulated data and GAW17 data, and conduct comprehensive comparison with some existing methods. Experimental results show that the proposed family-based method is powerful and robust for testing associations between multiple traits and rare variants, even if the data has some population stratification effect.

Introduction

With the launching of the Human Genome Project and the International Hapmap Project, scientific research on human diseases has been revolutionized by comprehensively understanding the human genome and its variations. Studies show that inheritance and genetics are crucial factors for the overwhelming majority of diseases. Therefore, researchers have been continuously exploring the associations between complex traits like human diseases and genetic variants in recent years (Risch & Merikangas, 1996). In the recent decades, a large number of common variants have been successfully identified by statistical methods, and some complex diseases are considered as the outcome of coaction of common variants and external environments (Hsu et al., 2012; Lin et al., 2013). However, researchers find that common variants only explain a small portion of heritability, which leads to a phenomenon of missing heritability (Eichler et al., 2010; Blanco-Gómez et al., 2016). In fact, rare variants (RVs) also contribute to disease heritability, hence recently lots of work has been done about RV association studies (Lin et al., 2015; Zhou et al., 2016; Wang, Ma & Zhou, 2017; Guo & Zhou, 2019).

Complex traits are usually recorded as multiple phenotypes, such as quantitative traits, binary traits, and ordinal traits, etc. Traditional genome-wide association studies (GWAS) focus on only one genetic variant and one trait at a time, which are not applicable to deal with the gene mapping problem of multiple traits (MTs) (Tong, Sun & Zhou, 2018). Therefore, some multiple-trait association tests (including linkage and association methods) have been developed. For example, based on idea of multiple-interval mapping, Joehanes (2009) proposed a multiple-trait multi-interval mapping method called MTMIM; Zhu & Zhang (2012) developed a nonparametric regression method for multiple longitudinal phenotypes based on multivariate adaptive splines; Zhou et al. (2016) proposed a nonparametric method that can be used to test for associations between MTs and RVs; Kwak & Pan (2016) proposed gene- and pathway-based association tests for MTs using GWAS summary statistics; Gai & Eskin (2018) gave a meta-analysis method for multiple-trait association analysis. Under the framework of linkage analysis, Tong, Sun & Zhou (2018) proposed an approach to simultaneously estimate QTL parameters for mapping MTs, which showed higher precision.

Although the above-mentioned methods can deal with the mapping problem of MTs in a certain degree, these methods focus on independent individuals. Family members have more chances to share same causal mutations, so the genetic heterogeneity is relatively small in family data, and it is a good strategy to test for the associations between genes/variants and MTs based on family design (Laird, Horvath & Xu, 2000). To date, there has been increasing evidence showing that family-based design performs better than population-based design in association studies of human complex diseases (especially for RVs) (Mathieson & McVean, 2012; Li et al., 2018). Hsieh, Chang & Tai (2014) proposed a maximum statistic test method based on family data (called MAX-FAM) to conduct multivariate trait association test. In addition, it is worth mentioning that investigation results have shown association studies based family data can effectively overcome the impact of population stratification on gene detecting (Fang, Sha & Zhang, 2012; Ma et al., 2013; Sha & Zhang, 2014; Guo & Zhou, 2019).

In this article, we propose a nonparametric test statistic based on the generalized Kendall’s τ theory to detect the associations between MTs and RVs using family data, and consider the statistical property of the new statistic. The new method has certain adjusting function to correct the effect of population stratification. Extensive simulation studies are carried out to illustrate the performance and robustness of the proposed method. The GAW17 data set is also applied to demonstrate its applicability.

Material and Methods

Data of n nuclear families of trios are considered, and suppose q traits are of interest. Denote Ti = (DTi, MTi, OTi)T as the q trait values of the ith family, where DTi = (DTi(1), DTi(2), ⋅⋅⋅, DTi(q))T, MTi = (MTi(1), MTi(2), ⋅⋅⋅, MTi(q))T and OTi = (OTi(1), OTi(2), ⋅⋅⋅, OTi(q))T represent the trait value of father, mother and child of the ith family, respectively. The traits considered here can be binary, ordinal or quantitative. Each subject has been genotyped at m RVs in a genomic region. Denote Gi = (DGi, MGi, OGi)T as the genotypic score matrix at the m RVs of the ith family, where DGi = (DGi1, DGi2, …, DGim)T, MGi = (MGi1, MGi2, …, MGim)T, and OGi = (OGi1, OGi2, …, OGim)T represent the genotypic score vector of father, mother and child of the ith family, respectively. DGil, MGil, OGil ∈ {0, 1, 2} are the number of minor alleles, and the frequency of minor allele is less than 5%. The main goal of this article is to test hypotheses as follows: H0: no association between MTs and RVs versus H1: Not H0.

The U-statistic and its distribution

Inspired by the idea of Zhou et al. (2016), we develop a new nonparametric method to test associations between MTs and multiple RVs based on family data. Firstly, we construct trait kernel function F(Ti, Tj) and genotype kernel function K(Gi, Gj) for family data, where F(Ti, Tj) is designed to measure the trait difference between families i and j, and K(Gi, Gj) is used to measure the genotype difference between families i and j (i, j=1, 2, …, n). Then based on the idea of generalized Kendall’s τ, we propose a U-statistic (1) U=n2−1 ∑i<jFTi,TjKGi,Gj.

Let Fij=FTi,Tj=FDTij,FMTij,FOTijT,

where FDTij=f1DTi1−DTj1,f2DTi2−DTj2,⋅⋅⋅,fqDTiq−DTjq,FMTij=f1MTi1−MTj1,f2MTi2−MTj2,⋅⋅⋅,fqMTiq−MTjq,FOTij=f1OTi1−OTj1,f2OTi2−OTj2,⋅⋅⋅,fqOTiq−OTjq.

If the kth trait is binary or quantitative, the function fk(⋅) is defined as fkDTik−DTjk=DTik−DTjk,fkMTik−MTjk=MTik−MTjk,k=1,2,…,qfkOTik−OTjk=OTik−OTjk.

If the kth trait ordinal, then fkDTik−DTjk=signDTik−DTjk=1,DTik>DTjk,0,DTik=DTjk,−1,DTik<DTjk,

fkMTik−MTjk=signMTik−MTjk=1,MTik>MTjk,0,MTik=MTjk,−1,MTik<MTjk,

fkOTik−OTjk=signOTik−OTjk=1,OTik>OTjk,0,OTik=OTjk,−1,OTik<OTjk,

where k = 1, 2, …, q.

We define (2) KGi,Gj=KDGi,DGj+KMGi,MGj+KOGi,OGj,

where KDGi,DGj= ∑l=1mKlDGil,DGjl is the kernel function for father genotypes measuring the genotype difference between the fathers in families i and j, and KlDGil,DGjl=lognDGilnDGjl, nDGil represents the total number of observed genotype Gil for all n fathers at variant l, and nDGjl has an analogous explanation. Similarly, kernel function for mother genotypes KMGi,MGj= ∑l=1mKlMGil,MGjl, and KlMGil,MGjl=lognMGilnMGjl; kernel function for child genotypes KOGi,OGj= ∑l=1mKlOGil,OGjl, and KlOGil,OGjl=lognOGilnOGjl. Notations nMGil, nMGjl, nOGil, nOGjl also have analogous explanations. The kernel functions are beneficial to integrate the weak signals from each RV.

Secondly, in order to calculate the U-statistic from the sample and construct an overall association test statistic based on family data to test the associations between MTs and RVs, we give the following proposition for the U-statistic.

Proposition: Based on the above definitions of kernel functions, the U-statistic can be simplified as U=2n−1∑l=1m ∑i=1nF ¯ilognDGilnMGilnOGil,

where F ¯i=1n ∑j=1nFij.

Proof: Replace the kernel functions F(Ti, Tj) and K(Gi, Gj) in formula (1), then substitute the detailed expressions of K(DGil, DGjl), K(MGil, MGjl) and K(OGil, OGjl) into formula (2). The U-statistic can be written by

U=n2−1∑i<jFTi,TjKGi,Gj=n2−1∑i<jFijKDGi,DGj+KMGi,MGj+KOGi,OGj=n2−1∑i<jFij∑l=1mKlDGil,DGjl+∑l=1mKlMGil,MGjl+∑l=1mKlOGil,OGjl=n2−1∑i<jFij ∑l=1mKlDGil,DGjl+KlMGil,MGjl+KlOGil,OGjl=n2−1∑i<jFij ∑l=1mlognDGilnDGjl+lognMGilnMGjl+lognOGilnOGjl=n2−1∑i<jFij ∑l=1mlognDGilnMGilnOGilnDGjlnMGjlnOGjl=n2−1∑i<jFij ∑l=1mlognDGilnMGilnOGil−lognDGjlnMGjlnOGjl=n2−1 ∑l=1mF12lognDG1lnMG1lnOG1l−lognDG2lnMG2lnOG2l+F13lognDG1lnMG1lnOG1l−lognDG3lnMG3lnOG3l+......+F1nlognDG1lnMG1lnOG1l−lognDGnlnMGnlnOGnl+......+Fn−1,nlognDGn−1,lnMGn−1,lnOGn−1,l−lognDGnlnMGnlnOGnl.

According to the definition of Fij (i, j=1, 2, …, n), it is obvious that Fij =  − Fji, Fii = 0. Therefore, U=n2−1 ∑l=1mF12+F13+...+F1nlognDG1lnMG1lnOG1l+F21+F23+...+F2nlognDG2lnMG2lnOG2l+......+Fn1+Fn2+...+Fn,n−1lognDGnlnMGnlnOGnl=n2−1 ∑l=1m ∑i=1nlognDGilnMGilnOGil∑j=1nFij.

Let F ¯i=1n∑j=1nFij,

then consequently, U=2n−1∑l=1m ∑i=1nF ¯ilognDGilnMGilnOGil.

Finally, we give the overall test statistic based on the above U-statistic to test association between MTs ad multiple RVs in family-based data. χFAM2=U−EU|TTVar−1U|TU−EU|T,

where T = (T1, T2, …, Tn), E(U|T) and Var(U|T) are the conditional expectation and variance of the U-statistic given T, respectively. Let pl be the minor allele frequency (MAF) of the lth RV, then under the null hypothesis and the assumption of Hardy-Weinberg Equilibrium, the conditional expectation and variance of the U-statistic given T have the following forms EU|T=2n−1 ∑i=1nF ¯i ∑l=1mElognDGilnMGilnOGil|T,

and VarU|T=2n−12 ∑i=1nFi ¯Fi ¯T ∑l=1mVarlognDGilnMGilnOGil|T,

where ElognDGilnMGilnOGil|T=3logn+21−pllog1−pl+2pllogpl+2pl1−pllog2,

and VarlognDGilnMGilnOGil|T=4pl1−pl210−7pllog21−pl+2pl1−pl14pl2−14pl+5log22pl1−pl+4pl21−pl7pl+3log2pl+8pl1−pl27pl−5log1−pllog2pl1−pl+8pl21−pl2−7pllogpllog2pl1−pl−56pl21−pl2logpllog1−pl.

The detailed derivation of E(U|T) and Var(U|T) are given in the Appendix. Zhang, Liu & Wang (2010) and Zhou et al. (2016) derived the distributions of some association statistics based on U-statistics. Under the null hypothesis, the test statistic χFAM2 follows an asymptotic chi-square distribution under the assumption of (approximate) independence of RV loci. The degrees of freedom p of χFAM2 is equal to the rank of the conditional variance matrix, i.e., p = rank(Var−1(U|T)). For convenience, the proposed nonparametric method based on family data for multiple-trait association testing is referred to as “Nonp-FAM”.

Simulation studies

Simulation design

We design a variety of simulation settings to evaluate the performance of the new proposed method (Nonp-FAM). At the same time, we compare Nonp-FAM with another competing method MAX-FAM (Hsieh, Chang & Tai, 2014) that is also based on family data. Besides, for comparison from data structure, we consider two commonly used methods that are based on independent individuals: CAST (Morgenthaler & Thilly, 2007) and SKAT (Wu et al., 2011).

Sample size: For the methods based on independent individuals, we set the sample size to be 600. For family data of trios, 200 independent families are considered.

RVs and genotypes: Assume that the number of RVs in the genetic region we considered is m (= 20 and 40) for each individual. For methods based on independent individuals, according to Hardy-Weinberg equilibrium, the frequencies of genotypes (AA, Aa and aa) at locus l are (1 − pl)2, 2pl(1 − pl) and pl2, respectively, where pl is drawn from U(0.03, 0.05), l = 1, 2, ⋅⋅⋅, m. Then the genotype can be randomly generated according to the probability distribution. Based on the generated genotype (AA, Aa and aa), the corresponding genotypic score Gil (=0, 1, 2) can be recorded. For family of trios, genotypes of the parents are randomly generated, and the genotypes of the offspring are generated by the transmission of the parents’ alleles.

Proportion of causal variants: Assumed that a total of 12 causal variants out of the 20 RVs, and 20 causal variants out of the 40 RVs are assigned. The specific form is denoted as: L′ = (1, 1, 0, 0, 1, 1, 1, 1, 1, 0, 0, 0, 1, 1, 1, 1, 0, 1, 0, 0) if m = 20 and L′ = (1, 1, 0, 0, 1, 1, 1, 1, 1, 0),  (0, 0, 1, 1, 1, 1, 0, 1, 0, 0, 1, 0, 1, 0, 1, 1, 1, 0, 0, 0, 0, 0, 0, 0, 0, 1, 1, 0, 0, 1) if m = 40, where the element 1 indicates that the corresponding locus is causal variant, and 0 indicates that it is non-causal variant.

Traits: Two traits are considered in the simulation studies. First, two-dimension continuous traits Ti=Ti1,Ti2T of each individual are generated according to the model Tik=μ+γkTGi+ɛik, k = 1, 2, where Gi = (Gi1, ⋅⋅⋅, Gim). Set the parameters μ = 0, γk = βk × L, where βk represents the risk factor, take β1 = 0, 0.2, 0.4, 0.6, 0.8, β2=12β1, and (ɛi1, ɛi2)T ∼ N(0, Σ), where Σ = 10.250.251. Second, the continuous traits can be divided into binary traits or ordinal traits according to certain percentages. Three schemes are designed here, namely, both of two traits are binary, both of two traits are ordinal, and mixture form of binary and ordinal traits. For the two binary traits, we take the percentile of one trait as 30% and the percentile of the other trait as 50%. Let the binary trait value be 0 if the continuous trait value is less than the assigned percentile, otherwise, it is 1. For the two ordinal traits, we set one of the traits into three categories dividing by the 50% and 67% sample percentiles, and the other trait includes four categories dividing by the 33%, 54% and 75% sample percentiles, respectively. For the mixed traits, we use the 40% sample percentile to generate the binary traits, and the generation of ordinal traits is the same as the one of the above three categories.

Significance level: The significance level α = 0.05 is considered in the simulation studies.

Simulation results

After generating the simulation data, we evaluate the estimated type I error rates and powers for all the above-mentioned four methods by 1000 replications. We compare the performance of these methods in each simulation scenario.

Evaluation on type I error rates

Table 1 shows the estimated type I error rates of the four methods at significance level of 0.05 in different trait design schemes. For 1,000 repetitions, the 95% confidence interval for the nominal level α is given by α−1.96α1−α1000,α+1.96α1−α1000. So the 95% confidence interval when α = 0.05 is (0.0365, 0.0635). In Table 1, we can see that Nonp-FAM, CAST, and SKAT can control the type I error rates very well in different scenarios. However, MAX-FAM slightly inflate the type I error rates in Scheme 2. So, we can conclude that the proposed Nonp-FAM approach is a valid test.

Table 1 Estimated type I errors of the four methods based on designed parameters.

Test		20RVs			40RVs		
	Scheme 1	Scheme 2	Scheme 3	Scheme 1	Scheme 2	Scheme 3	
Nonp-FAM	0.0440	0.0450	0.0430	0.0470	0.0440	0.0380	
MAX-FAM	0.0400	0.0650	0.0600	0.0490	0.7100	0.0550	
CAST	0.0480	0.0420	0.0440	0.0390	0.0430	0.0450	
SKAT	0.0440	0.0440	0.0460	0.0410	0.0510	0.0370	
Notes.

Note Scheme 1 two binary traits

Scheme 2 two ordinal traits

Scheme 3 mixture of binary and ordinal traits

The significance level α = 0.05 in the simulations.

Power comparisons

In order to better demonstrate the advantages of the Nonp-FAM method in detecting the associations between MTs and RVs, we design a variety of simulation settings to compare the powers for the four methods. The simulation results of power comparison for the four methods when the traits are two binary, two ordinal, and mixture of binary and ordinal traits are listed in Figs. 1 to 3, respectively.

Figure 1 Power comparisons of the four methods for two binary traits in Scheme 1.

Figure 2 Power comparisons of the four methods for two ordinal traits in Scheme 2.

Figure 3 Power comparisons of the four methods for a mixture of binary and ordinal traits in Scheme 3.

We can draw several conclusions from the Figs. 1 to 3. On one hand, the powers of Nonp-FAM are much higher than those of MAX-FAM even though both are based on family data of trios. On the other hand, the powers of Nonp-FAM are higher than those of CAST and SKAT that are based on independent individuals. Therefore, for the RV association analysis, the Nonp-FAM method relatively outperforms the other three methods whether the traits are binary, ordinal, or the mixture of them. Besides, as expected, with the increase of risk factors (β1 and β2), the powers of all methods show increasing tendency. This is because higher risk factors correspond to higher heritability. The above experimental results show that the proposed Nonp-FAM method performs effectively when the sample size is 200 independent families. To verify the performance of the proposed method with different sample sizes, the simulations with a sample size of 120 independent families were additionally conducted, and the results are shown in Figs. S1 to S3 of this article.

Family-based association studies are particularly valuable because of the consideration of family history for the disease of interest, and population stratification is still a big concern in population-based association studies. Here, we design an additional experiment to demonstrate that the Nonp-FAM method has the advantages in analyzing data with population stratification. In the new simulation, we consider 100 families of trios with population stratification effect consists of three parts: 30 families whose RV frequencies are randomly drawn from interval (0.03, 0.037), 30 families whose RV frequencies are drawn from (0.037, 0.044), and 40 families whose RV frequencies are drawn from (0.044, 0.05). Meanwhile, we consider two traits which are generated similar to Scheme 3, i.e., the mixture of binary and ordinal traits. The parameters to generate data are same as those in the previous simulation design. We calculate the powers of Nonp-FAM under the simulated data with population stratification and without population stratification (RV frequencies are drawn from the same interval (0.03, 0.05) for 100 families). The comparison results are shown in Table 2.

Table 2 Power comparisons of the Nonp-FAM method with and without population stratification effect.

RVs	Stratification	β1= 0.8	β1= 0.6	β1= 0.4	β1= 0.2	
		β2= 0.4	β2= 0.3	β2= 0.2	β2= 0.1	
20	No	0.9540	0.8470	0.5080	0.1400	
(12 Causal)	Yes	0.9720	0.9000	0.5970	0.1880	
40	No	0.9690	0.8970	0.6230	0.1930	
(20 Causal)	Yes	0.9940	0.9490	0.7820	0.2920	

As we can see from Table 2, the Nonp-FAM method has a better performance when there is underlying population stratification in the data structure. Therefore, Nonp-FAM is an executable and effective approach to analyze family data especially in the presence of population stratification.

Furthermore, we conducted extended simulation experiments with three ordinal traits, in which the causal RVs only affect the first two traits and there is no genetic association between the RVs and trait 3. The generating process of simulated data is omitted here. We performed 1000 replications to calculate the powers of two family-based methods (the Nonp-FAM method and the MAX-FAM method) at the significance level of 0.05, and the simulation results were listed in Table S1 in the Supplemental Material of this article. From this table, we can find that our new method is also effective when the RVs only affect some of the traits, and the Nonp-FAM method is still more powerful than the MAX-FAM method by comparing the corresponding powers.

Real data analysis

To demonstrate the application of the Nonp-FAM method, we apply the method to the Mini-Exome genotype data provided by the genetic analysis workshop 17 (GAW17) (https://www.gaworkshop.org/). The GAW17 data set consists of real sequence information on a large number of genes from the 1000 Genomes Project (http://www.1000genomes.org) and simulated phenotype data. There are two kinds of data sets: one is 697 unrelated-individual data genotyped at 24,487 autosomal SNPs in 3,205 genes and the other is family data distributed in 8 extended families with the same sample size. The range of MAFs of common and rare variants is from 0.07% to 25.8%. As an example, we consider three related quantitative risk traits Q1, Q2, and Q4 in the data set as the MTs. The GAW17 reported that traits Q1 and Q2 were influenced by nine and 13 genes, respectively. Here we select the ARNT gene with 2236 variant loci, which is located on Chromosome 1 and has significant main effect on trait Q1 (Almasy et al., 2011).

Next, we use the proposed method to detect associations between the MTs and RVs. The eight extended families can be regarded as eight subpopulations. For each subpopulation, we organized the relevant individual data into trio family data. Thus 194 trio families was extracted. In addition, 593 non-zero and RV loci (MAF <0.01) were further selected from the 2,236 variant loci. Forty loci were composed as a genetic region to be tested at one time (i.e., regions 1–40, 41–80, ⋯ , and 561–593). The causal SNPs C1S6533 and C1S6540 in the ARNT gene (Almasy et al., 2011) are in the assigned region 361–400 in this example. We calculate the pair-wise linkage disequilibrium coefficients among these considered RV loci, and there is only weak linkage disequilibrium between these loci, so these RV loci are approximately independent. Then the Nonp-FAM method is used to detect the associations between the MTs and each region of RVs. The values of the test statistic and the corresponding P-values of the Nonp-FAM method are listed in Table 3. From the testing results, we can see that five significant regions were detected at the significance level of 0.05, where one significant region was detected at the significance level of 0.01. The P-value for testing the region 361–400 is 0.0152, which means the region containing the causal SNPs C1S6533 and C1S6540 is successfully detected by the Nonp-FAM method.

Table 3 The values of the test statistic and the P-values of the Nonp-FAM method at the significance level of 0.05.

Region	χ2FAM	P-value		Region	χ2FAM	P-value	
1–40	11.8135	0.2240		321–360	7.8182	0.5526	
41–80	10.9650	0.2815		361–400	20.4716	0.0152	
81–120	27.0099	0.0014		401–440	13.4568	0.1403	
121–160	18.3693	0.0311		441–480	3.0737	0.9613	
161–200	15.8190	0.0708		481–520	21.1546	0.0120	
201–240	16.2765	0.0613		521–560	6.2609	0.7135	
241–280	21.3409	0.0112		561–593	7.5594	0.5791	
281–320	12.5995	0.1816		–	–	–	

Discussion

In the recent decades, a vast majority of genetic variants have been identified to be associated with human complex traits, and RVs play a vital role in the association studies. Also, considering MTs in genetic association studies can boost statistical power. In this article, based on the idea of generalized Kendall’s τ, we construct the U-statistic using family data and then build the overall the test statistic χFAM2. The associations between MTs and RVs can be effectively detected by the proposed non-parametric method Nonp-FAM. Through extensive simulation experiments and GAW17 data analysis, we illustrate the effectiveness and feasibility of the Nonp-FAM method. The numerical results show that the new method outperforms an existing family-based association testing method, as well as two commonly used population-based association testing methods (one burden test and one nonburden test). It is also worth mentioning that the new method has high power for the family data with population stratification, which sufficiently shows the advantage of using family-based method in association analysis.

In fact, the proposed method can be used to analyze family data with other structures, not only the trio structure in this article. The key of the new approach is constructing ideal trait kernel functions and genotype kernel functions between two families. This method can be extended to fit other family structures by adding information of other children in the correspond kernel functions. However, more general methods to construct kernel functions still need to be considered to face the association studies of pedigree data whose structure is more complex.

The new method is a non-parametric method, and it is robust to any types of MTs. The dimension of MTs has certain impact on the calculation of the test statistic χFAM2, but it will not impact the power of association testing. Besides, the number of RV loci has little impact on our method, since the genotype kernel function has the additive form about paired family members and the considered loci.

In our method, we mainly consider the situation that all the RV loci are supposed to be independent. In practice, some loci are dependent and some loci may have interaction with certain environmental factors, and these conditions can be added in the theoretical part of our method. In fact, before using the Nonp-FAM method, all pair-wise linkage disequilibrium coefficients can be calculated first to verify the assumption of independence, like we did in our example. If some loci are not independent, we suggest collapsing the dependent RVs into ‘super variant’ by weighting the genotypes of these loci, so that the rest RVs will be independent of the ‘super variant’, then the proposed method can be used. The distribution of the proposed statistic χFAM2 was obtained under the condition of large sample size, so the test power may decrease if the sample size is not large. For higher heritability, the power of the new method can achieve about 0.9 when considering 100 trios (sample size is 300) at the significance level of 0.05; and for middle heritability, the power of it reach about 0.9 when taking 200 trios into account (sample size is 600). Besides, the effect directions of RVs may also have certain impact on the new method. In our future work, we will continue to consider these issues.

Supplemental Information

Supplemental Information 1 The data used for analysis.

Click here for additional data file.

Supplemental Information 2 The code used in simulation and data analysis.

Click here for additional data file.

Supplemental Information 3 Supplemental table and figures.

Click here for additional data file.

Supplemental Information 4 Appendix.

Click here for additional data file.

The authors would like to thank the joint Editor and referees for helpful comments that greatly improved the presentation of the article. We also thank the Genetic Analysis Workshop 17 for providing the data set analyzed in this article.

Additional Information and Declarations

Competing Interests

Author Contributions

Data Availability

The authors declare there are no competing interests.

Jinling Chi performed the experiments, analyzed the data, prepared figures and/or tables, and approved the final draft.

Meijuan Xu analyzed the data, prepared figures and/or tables, and approved the final draft.

Xiaona Sheng performed the experiments, prepared figures and/or tables, wrote the code, and approved the final draft.

Ying Zhou conceived and designed the experiments, authored or reviewed drafts of the article, and approved the final draft.

The following information was supplied regarding data availability:

The data and code are available Supplemental Files.

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
