# Peer review of "Association detection between multiple traits and rare variants based on family data via a nonparametric method"

_PeerJ, doi:10.7717/peerj.16040_

## Round 0.1 · original submission · Major Revisions

The manuscript needs major revisions as stated by the referees. Please exhaustively address all issues raised.

·

Basic reporting

no comment

Experimental design

no comment

Validity of the findings

no comment

Additional comments

Based on the generalized Kendal’s tau theory, the authors proposed a non-parametric statistic Nonp-FAM to test for the association between multiple traits and multiple rare variants using a series of family trio data. The asymptotic distribution of the test statistic under the null hypothesis is given. Simulation studies and real data analysis are present to exhibit the advantage of the proposed method. I have the following comments. (1) Whether Nonp-FAM works if there exist common variants, and what adjustment should it do? (2) In the case of multiple traits, if we know some are important and some are not much important, how can you incorporate this prior knowledge into the method? (3) In equation (2) of the kernel function of two families, as the genotype of child depends heavily on his/her parents’, why the third term K(OG_i, OG_j) shares the same weight as the previous two terms? (4) In line 126, is the rank of Var(U|T) same or not for every replication in the simulation study? (The observed data should be different for each replicate). (5) Are CAST and SKAT in lines 134-135 applicable to family data? (6) If there exist father/mother-child pair data, can you utilize these in the proposed statistic? (7) In line 287, you claim the proposed method can correct the impact of population stratification. It is hard for the reader to find the numerical support in the simulation results.

Reviewer 2 ·

Basic reporting

The new analytical model proposed by the authors looks like it could solve the problems that low frequency SNPs have had in previous analyses. However, there are some problems with the validation design and the method validation is relatively crude.

Experimental design

1. Sample size: For the methods based on independent individuals, we set the sample size to 600. and for family data of trios, 200 independent families are considered. considered.

The selection of sample size in the article is based on a relatively small single sample size, which is in a small class in GWAS studies. With this sample size, the model described by the authors has a relatively ideal p-value. However, comparisons of models with different sample sizes should be done, such as 50:50... .300:300... .1000:1000... .20,000:20,000.
2. TDT analysis is the classical study regarding trios, and a comparison between the low frequency mutations in TDT analysis and the authors' method should be added.
3.The authors' model appears to be effective in a selected segment, but does it achieve efficacy in all of the genome-wide data. The results of the analysis in the genome-wide SNP data should be carried out.

Validity of the findings

The conclusions appear relatively reliable

---

## Round 0.2 · accepted · Accept

The authors have adequately addressed all issues raised.

The Section Editor noted that Data Availability statement needs to have fuller information on the source dataset and how to download it.